# SEMANTIC WORLD MODELS

## ABSTRACT

Planning with world models offers a powerful paradigm for robotic control. Conventional approaches train a model to predict future frames conditioned on current frames and actions, which can then be used for planning. However, the objective of predicting future pixels is often at odds with the actual planning objective; strong pixel reconstruction does not always correlate with good planning decisions. We posit that instead of reconstructing future frames as pixels, world models only need to predict task-relevant *semantic* information about the future. To do this, we pose world modeling as a visual question answering problem, about semantic information in *future frames*. This perspective allows world modeling to be approached with the same tools underlying vision language models. We show how vision language models can be trained as "semantic world models" through a supervised finetuning process on image-action-text data, enabling planning for decision-making while inheriting many of the generalization and robustness properties from the pretrained vision-language models. We demonstrate how such a semantic world model can be used for policy improvement on open-ended robotics tasks, leading to significant generalization improvements over typical paradigms of reconstruction-based action-conditional world modeling.

## 1 INTRODUCTION

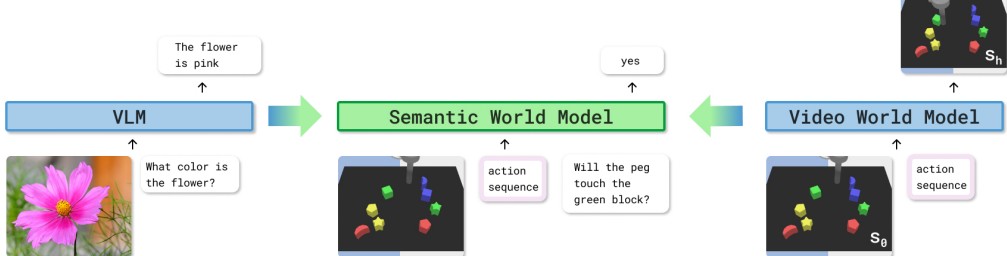

Figure 1: Comparison between Vision-Language Models, Video World Models, and Semantic World Models. While Vision-Language Models answer questions about static observations and Video World Models predict future observations given actions, Semantic World Models take observations and actions as input to directly answer questions about the future outcomes of those actions.

World models are a class of learning methods capable of absorbing large amounts of data to make generative predictions about future outcomes in the world. These predictions can then be used to inform decision-making via planning (Williams et al., 2016; Hafner et al., 2019; Rybkin et al., 2021; Hansen et al., 2022), helping policies acquire generalizable and robust behaviors. The practical instantiations of world models are diverse, ranging from smaller state-based dynamics models (Ai et al., 2025) to large action-conditioned video prediction models (Ball et al., 2025). Across these instantiations, pixel-level reconstruction of future observations is commonly used as a training recipe. While these approaches are often successful at generating realistic images, as evident from high-quality video generations, they can be challenging to use for planning. Despite the visual fidelity, these predictions often miss (or misrepresent) key semantic details necessary for decision making, e.g., the details of precise dexterous contact. While there have been suggestions for modeling "task-relevant" latent representations (Zhang et al., 2021; Hansen et al., 2022; Zhu et al., 2023), these

methods often impose additional assumptions on the availability of rewards (Hansen et al., 2024) or known factors (Locatello et al., 2020), making them challenging to use in practice across a variety of world modeling problems.

If pixels are not necessary for planning, what is actually needed to make decisions about acting in the world? We posit that the ability to predict *semantic* information about future outcomes is sufficient. Rather than forecasting raw visual frames, world models should capture task-relevant information about objects and their interactions, e.g., "Did the arm get closer to the object?", "Did the red cube tip over?", "Was the blue moon picked up?". In this work, we frame such information as a visual question-answering (VQA) problem about the future, leveraging the fact that any desired outcome can be expressed as a set of yes/no questions[1]. That is, *the problem of world modeling can be redefined as a VQA problem about outcomes in the future*.

There already exists a class of models with extensive tooling for VQA on static observations, i.e., vision-language models (VLMs). For world modeling, VLMs offer two key advantages: they provide a strong foundation for VQA through large-scale pretraining and broad generalization, and they encode prior knowledge about which tasks and semantic features are relevant in a scene. These strengths make frontier VLMs well suited to formulating task-relevant questions and producing reliable answers when given static observations. However, their lack of predictive capacity about future outcomes limits their direct utility for decision-making.

This work introduces the paradigm of Semantic World Model (SWM) – a generalizable world model that is represented as an action-conditional vision-language model that answers questions about the semantic effects of actions in the future. Unlike traditional world models that predict future frames, a Semantic World Model *answers questions about the future* given current observations (represented as an image) and a sequence of actions. As shown in Fig. 1, the model takes as input the current observations, a proposed action sequence, and a natural language query about the future. It then generates an answer by understanding the consequences of taking the actions in the environment. Since SWM is fundamentally a task-agnostic world model, it can be trained on general sequential play and suboptimal data with minimal assumptions for data quality. The training data can be easily obtained from any (expert or non-expert) data corpus in the format of current observations, actions, questions (about the future), and expected answers.

The ability to reason about outcomes in the future with an SWM enables flexible open-world multi-task planning in action space: given a task specification in natural language, we could leverage a pre-trained frontier VLM (OpenAI, 2024; Beyer et al., 2024) to decompose the task specification into a set of questions and expected answers in text form. Given this QA set, SWM can then be used to plan actions that elicit the expected answers to these questions *in the future* with high likelihood. While a plethora of techniques can be used for this planning, in this work we show compatibility with both zero-order sampling-based methods (Rubinstein & Kroese, 2004; Williams et al., 2016) and first-order gradient planning methods (Ruder, 2017; Rybkin et al., 2021) that perform optimization with respect to the expected likelihood objective. We show that these planning methods can be made computationally tractable, enabling significant test-time improvement over nominal action selection methods. Moreover, we demonstrate the extensibility of such planning methods to multi-step long-horizon problems.

We empirically evaluate SWM on a suite of multiple different tasks in two commonly used multi-task simulation domains - Language Table (LangTable) (Lynch et al., 2022) and OGBench (Park et al., 2025). We show that (1) SWM can accurately answer questions about future outcomes while generalizing to novel scenes, and (2) SWM can be combined with standard sampling-based planning techniques and a gradient-based improvement technique to solve diverse robotics tasks with considerable policy improvement through test-time optimization. Through SWM, we introduce a new class of world models that leverage the rich pretraining knowledge from VLMs for grounded, flexible, and scalable robotic control.

## 2 RELATED WORK

**Vision-Language Models (VLMs)** broadly encompass representation learning methods and multimodal generative models trained on vision and language data. Representation learning methods

---

[1] other question-answer types may be applicable as well

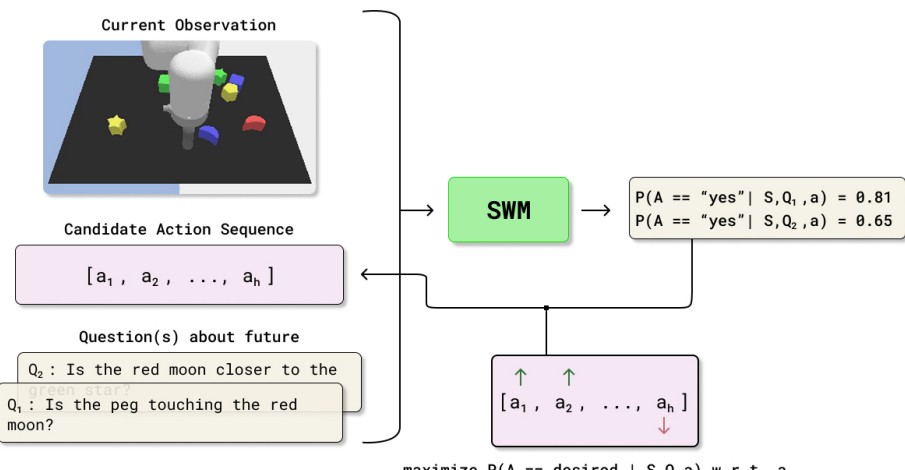

Figure 2: **Overview of Semantic World Models.** SWM is a VLM adapted to answer questions about the future after executing the actions from the current state. By querying the model with actions and question about the future, the model can evaluate the fitness of each action sequence using the desired answers, and enable planning under the model.

jointly train a vision encoder and a text encoder by aligning their encoded representations. These representations can then be utilized in various applications, such as classification, retrieval, and control. CLIP (Radford et al., 2021) learns such representations from image-text data by utilizing a contrastive loss, contrasting positive image-text pairs with negative pairs. SigLIP (Zhai et al., 2023) replaces the contrastive loss with a pairwise sigmoid loss to facilitate scalable training. Multi-modal generative models, commonly known as VLMs, enable a broad range of promptable behaviors such as understanding, summarizing, and question answering (OpenAI, 2024; Gemini Team, 2023; Deitke et al., 2024; Bai et al., 2023; Beyer et al., 2024; Touvron et al., 2023). A VLM takes in an image and a language prompt as input and generates a natural language response. They are typically trained with a next-token prediction objective. Recently, a family of vision-language-action models (VLAs) has been introduced to bring the vision-language understanding capabilities of VLMs to embodied decision-making (Brohan et al., 2023; Kim et al., 2025; Black et al., 2024). VLAs are trained on annotated robot trajectories to generate actions conditioned on image observations and language instructions. OpenVLA (Kim et al., 2025) directly predicts discrete action tokens, while Pi-0 (Black et al., 2024) decodes actions via a diffusion action head. Unlike VLAs, an SWM takes in observations, actions, and a natural language prompt as input, and generates a natural language response about the future after taking the actions. In some sense, an SWM can be viewed as an "inverted" VLA, where the actions become the input and the language becomes the output. We hypothesize that using language as the output format can better retain the pretraining knowledge of VLMs, since they were trained with next token prediction objectives.

**World Models for Control** are approximate models of the dynamics of the world, typically trained to predict future observations conditioned on current observations and actions. The ability to forecast the future without interacting with the world can greatly facilitate decision-making and control. A prominent line of work focuses on planning with world models. (Chua et al., 2018; Hafner et al., 2019; Rybkin et al., 2021). PETS (Chua et al., 2018) learns a one-step dynamics model and applies the cross-entropy method to plan for optimal actions for a given reward. PlaNet (Hafner et al., 2019) learns a recurrent latent dynamics model with a reconstruction objective and applies planning in the latent space. LatCo (Rybkin et al., 2021) leverages collocation-based planning to enable long-horizon planning with latent dynamics models. Another line of work utilizes world models as a simulator for reinforcement learning (Hafner et al., 2020; Zhang et al., 2021; Hansen et al., 2022). Dreamer (Hafner et al., 2020) and TD-MPC (Hansen et al., 2022) use a latent dynamics model to generate rollouts for actor-critic policy optimization, achieving remarkable sample efficiency. (Zhang et al., 2021) learns a latent representation predictive of dynamics and reward, which can then be used as an invariant representation for RL policies. Recently, world models have

been used together with imitation learning methods to facilitate out-of-distribution generalization (Du et al., 2023; Zhu et al., 2025). UniPi (Du et al., 2023) uses a world model as a high-level planner to condition low-level policies. UWM (Zhu et al., 2025) trains a unified video-action diffusion model, incorporating video data into pretraining to improve generalization. Unlike these explicit world models, SWM understands the dynamics of the world by reasoning in language space, allowing the model to bootstrap from the Internet-scale pretraining of VLMs. SWM can then be used with planning techniques to derive versatile language-conditioned policies. Additional work also explores abstractions closely connected to Semantic World Models. MEAD (GX-Chen et al., 2025) defines an abstract MDP over items and attributes to simplify exploration and modeling. VLWM (Chen et al., 2025) creates a VLM-based world model where, given a goal, it predicts both actions and how the state of the world changes after the actions are executed. Prior work on predicate learning (Silver et al., 2025; Athalye et al., 2025) learn abstract or semantic predicates to decompose long-horizon tasks into shorter subgoals, a direction that is complementary to SWM's use of future QA for planning.

## 3 METHOD

This section presents details of the data generation pipeline, the SWM architecture, and the training methodology. It then touches on the sampling-based and gradient-based planning methods used for policy extraction under SWM. Fig. 2 provides an overview of the model and planning procedure.

### 3.1 DATASET GENERATION

To train a world model to answer questions about the future, we generate a state-action-question-answer (SAQA) dataset defined as

$$\mathcal{D}_{\text{SAQA}} = \{(S_i, a_{i:j}, Q_{S_j}, A_{S_j}), \dots\} \quad \text{where } j = i+h$$

where $S_i$ represents the current state (RGB frame in our case), $h$ is the horizon, $a_{i:j}$ is a sequence of actions taken from state $S_i$, and $Q_{S_j}, A_{S_j}$ is a question answer tuple about the future state $S_j$ which is reached by taking actions $a_{i:j}$ from state $S_i$. Fig. 3 illustrates a single state paired with multiple questions and answers in the dataset.

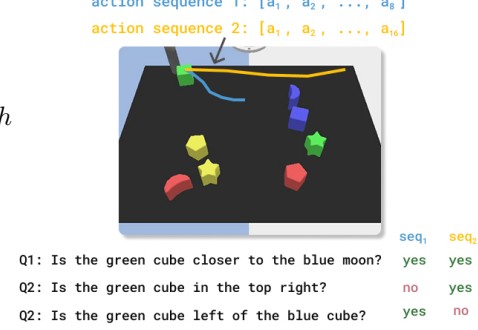

Figure 3: Example initial state in the SAQA dataset with two action horizons and six question-answer pairs.

We generate the SAQA dataset from a dataset of trajectories $\{T_1, T_2, \dots\}$, where each trajectory is given by a sequence of state-action tuples $\{(S_0, a_0), (S_1, a_1), \dots\}$. In our case, each state comprises an image observation and privileged information, such as object positions, which we use for programmatic question generation. For each state $S_i$ in the trajectory, we sample multiple different action horizons $h$. As shown in Fig. 3, for each sampled horizon $h$, we use the oracle information from future state $S_{i+h}$ to create a set of questions and answers, giving us the final dataset to train our model. For each type of question generation, we include multiple phrasings in our training dataset. Examples of the question types for training and the reward for each task are provided in section A.3.2.

### 3.2 SEMANTIC WORLD MODELS

We proceed to design a model capable of answering questions about future events conditioned on actions. A model with such capability is fundamentally a visual question-answering model with action conditioning. Therefore, it is natural to bootstrap from large pretrained VLMs to transfer their generalization capabilities to robotics tasks. We base our SWM architecture on an open-source VLM, PaliGemma (Beyer et al., 2024). The model contains three core pretrained components: a transformer-based autoregressive language model with a token embedding size $d_{\text{tok}}$, a vision encoder $v_\phi$ with a feature size $d_{\text{img}}$, and a projection matrix $W \in \mathbb{R}^{d_{\text{tok}} \times d_{\text{img}}}$. The PaliGemma architecture is based on the Gemma LLM (Gemma Team et al., 2024) and the SigCLIP vision encoder $V_{\text{sc}}$ (Zhai

et al., 2023). $W$ is used to project from $Z_{\text{sc}}$ to $Z_{\text{LLM}}$, where $Z_{\text{sc}}$ is the feature space of $v_\phi$, and $Z_{\text{LLM}}$ is the input token embedding space of the LLM. We use the 3B parameter checkpoint from PaliGemma as our base model.

To adapt the base model to answer questions about a specific future as a result of the actions, the model needs to be conditioned on these actions. To this end, we create a new projection matrix $P \in \mathbb{R}^{d_{\text{tok}} \times d_{\text{act}}}$ which projects a single action $a \in \mathbb{R}^{d_{\text{act}}}$ into the latent space $Z_{LLM}$ similar to the $W$ projection matrix. Given a tuple $(S_i, a_{i:j}, Q_{S_j}, A_{S_j})$ from the dataset $\mathcal{D}_{\text{SAQA}}$, we construct the input sequence by concatenating the image embeddings, action embeddings, and question token embeddings as $\texttt{concat}\left(W^\top V_{sc}(S_i), P^\top a_i, P^\top a_{i+1}, \ldots, P^\top a_j, Q_{S_j}\right)$. The model is then fine-tuned in an end-to-end manner to predict the target answer $A_{S_j}$ by optimizing the standard cross-entropy loss

$$\mathcal{L} = -\log p(A_{S_j} | S_i, a_{i:j}, Q_{S_j}).$$

This training procedure enables the model to capture the dynamics of the environment in language space to answer questions about future states without explicitly generating pixel-level representations.

## 3.3 Planning with Semantic World Models

Planning with world models requires evaluating the value of action sequences. For each task, we can define a set of questions (e.g., "is the gripper touching the block") and desired answers (e.g., "yes"). We can then derive a scalar score by combining the likelihood of the model generating the desired answer for each question, weighted by some heuristic weights. Specifically, each task is defined as a set of questions, answers, and weights $\mathcal{T} := \{(Q_i, A_i^*, W_i)\}_{i=1}^k$. Given an observation $S$ and a sequence of actions $a_{1:n}$, we calculate its value under the task as:

$$V^\mathcal{T}(S, a_{1:n}) = \sum_{i=0}^k W_i \cdot p_{\text{wm}}(A_i^* | S, a_{1:n}, Q_i) \tag{1}$$

We empirically find that rewarding the model for achieving the desired outcome earlier in the action sequence leads to better performance. To do so, we break each full action sequence down to sub-chunks of length $c$, and then query the model on action sequences with increasing numbers of concatenated sub-chunks:

$$V^{\mathcal{T},c}(S, a_{1:n}) = \sum_{i=0}^k \sum_{\substack{j=c \\ j+=c}}^n W_i \cdot p_{\text{wm}}(A_i^* | S, a_{1:j}, Q_i) \tag{2}$$

Setting $c = 1$ is equivalent to evaluating the model once for every single action in the sequence, and setting $c = k$ is equivalent to the vanilla formulation in Eqn. 2. With a well-defined value function, we can apply various planning techniques to extract optimal actions using the model.

### 3.3.1 Sampling-Based Planning

Sampling-based planning provides a straightforward approach to planning with the model. An example is Model Predictive Path Integral (MPPI) control algorithm Williams et al. (2016), which maintains a Gaussian distribution of action parameters and iteratively refines it by querying the model. The action distribution is initialized as $\mathbf{a}^{(0)} \sim \text{Unif}(a_{\min}, a_{\max})$. At each iteration, we sample a set of $K$ control sequences $\{\mathbf{a}^{(k)}\}_{k=1}^K$ from the current action distribution. The value of each of these sampled trajectories $V_k$ is computed using our SWM . The distribution for the next iteration is $\mathbf{a}_{t+1} \sim \mathcal{N}\left(\mu_t, \sigma_t^2\right)$ where

$$\mu_t = \sum_{k=1}^K \frac{\exp\left(\frac{V_k}{\lambda}\right)}{\sum_{j=1}^K \exp\left(\frac{V_j}{\lambda}\right)} \mathbf{a}_t^{(k)}, \qquad \sigma_t^2 = \sum_{k=1}^K \omega_k \left(\mathbf{a}_t^{(k)} - \mu_t\right)^2 \tag{3}$$

and $\lambda$ is a temperature parameter controlling exploration. For our rollouts, we execute the mean sequence of the last iteration.

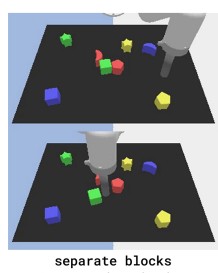 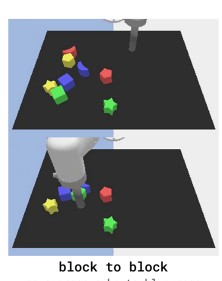 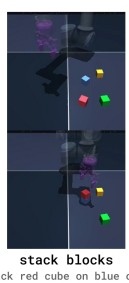 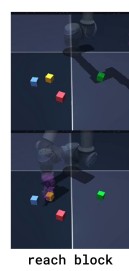

**peg to block**     **separate blocks**     **block to block**     **stack blocks**     **reach block**
peg to green star   separate green cube and red pentagon   move green cube to blue moon   stack red cube on blue cube   reach yellow cube

Figure 4: Examples of each evaluation task. The top frame represents the initialization, and the bottom frame represents task completion.

### 3.3.2 GRADIENT-BASED PLANNING

For more complicated tasks, sampling-based planning methods typically require a large number of samples and optimization iterations, which become increasingly hard to scale for a large model like SWM . To reduce the number of samples and model forward passes, we propose to use a gradient-based optimization procedure together with a base proposal policy. The gradient provides directed information for optimizing the model, thus converging faster than sampling-based techniques. The base proposal policy can effectively trim down the planning search space. Given a base policy $\pi_b$ and a control sequence $\mathbf{a} \sim \pi_b(S)$, and our semantic world model $p_{\text{wm}}$, we perform a gradient ascent to optimize the following objective:

$$J^{\mathcal{T}}(\mathbf{a}) = V^{\mathcal{T},c}(S, \mathbf{a}) \tag{4}$$

Where $\mathbf{a}$ is the control sequence we are optimizing over, $\mathcal{T} = \{(Q_i, A_i^*, W_i)\}_{i=1}^k$ is the list of questions, desired answers, and weights, $c$ is our reward subchunk size, and $S$ is our state. To improve the stability of our objective, we employed gradient norm clipping on the actions before each gradient step. Fig. 11 shows a visualization of this optimization process. Appendix A.5.8 compares the of planning times for each method.

### 3.4 MULTISTEP TASKS

To solve long-horizon tasks, we can extend the aforementioned planning procedure to a multi-step formulation. We leverage the capabilities of SWM to decide task progress and transition between subgoals without requiring any additional components. Concretely, we define a series of sequential subgoals $g_1, g_2, \ldots, g_T$, where each subgoal $g_t$ is associated with a question and a desired answer corresponding to when the subgoal was completed. We sequentially execute each subgoal and verify its completion using SWM. This is feasible at no additional cost because we include zero-horizon examples in the training dataset. For example, in the block picking task, we used the following sub-goals: ["Is the block grasped?", "Is the block stacked on top of the other block?"], with the desired answers ["yes", "yes"] in order to accomplish a two-stage task. This method is used to extend planning to multi-step LangTable tasks.

## 4 EXPERIMENTS AND RESULTS

### 4.1 EXPERIMENTAL SETUP

We evaluate SWM in two simulation environments, LangTable (Lynch et al., 2022) and OGBench (Park et al., 2025), capturing combinatorial generalization and dexterous manipulation. Fig. 4 shows examples of tasks in each domain. We provide an overview of the experiment setup in this section and defer the details to Sec. A.2

**LangTable** (Lynch et al., 2022) We evaluate our approach on *reaching*, *separating blocks*, and *pushing* in the LangTable environment, using both sampling-based planning and gradient-based improvement over a base policy. We train SWM on a mixture of expert data collected with a scripted policy and suboptimal data collected with a random policy. To evaluate in out-of-distribution con-

ditions, we change the block color combinations during evaluation to test compositional generalization. For example, our training data only includes the red pentagon, and we evaluate on a green pentagon and a novel purple pentagon.

**OGBench** (Park et al., 2025) We evaluate on *cube reaching* and a custom *cube stacking* task. We train SWM on a mixture of optimal and suboptimal data, collected using the provided noisy expert data and play data from OGBench, respectively. To measure generalization, we change the background color during evaluation.

For both environments, we use a per-task Diffusion Policy (Chi et al., 2023) trained on 300 expert trajectories for 100 epochs as the base policy. The expert trajectories were collected using the same experts as in the offline dataset.

One important aspect of training was ensuring the dataset was balanced in both the number of each possible question type and the answer distribution for each respective question. For example, for each state in the LangTable environment, there are $\binom{8}{2}$ possible questions about whether two blocks are touching, but 8 questions about whether the end effector is touching a given block. Similarly, most blocks are separated in the initial states of the LangTable environment, leading to far more 'yes' answers than 'no' answers. The imbalance is addressed during training by oversampling tuples such that there is a balanced amount of question types and answer distributions.

## 4.2 BASELINES

We compare Semantic World Models to the following baselines. Details about each baseline and hyperparameters are described in Sec. A.2

**IDQL** (Hansen-Estruch et al., 2023): IDQL is an offline RL baseline which uses IQL Kostrikov et al. (2022) to reweight the a behavior diffusion policy. For each task, we take the offline dataset used for our Semantic World Model and combine it with the per-task expert dataset used for the base policy. This combined dataset is labeled with binary rewards and used to train our IDQL policy. The architecture and hyperparameters of the diffusion policy used as the IDQL behavior policy are the same as for the base policies, except with a horizon of 8.

**Action Conditioned Video Diffusion (AVD)**: To compare against a pixel-based world model, we train an action-conditioned k-step video diffusion model. We model its architecture after the backbone used in Unified World Models (Zhu et al., 2025). Using this video diffusion model, we predict the future frame conditioned on the proposed action sequence and use the SWM model to perform VQA on this predicted frame, which we use as a reward for MPPI planning. The initial trajectory candidate samples are generated through our base diffusion policy.

## 4.3 RESULTS

Our evaluation aims to address the following questions: (1) Is SWM an effective world model for decision making? (2) Does suboptimal data improve modeling performance? (3) Does SWM preserve the generalization capabilities from the base VLM?

**Is SWM an effective world model for decision making?**
To evaluate the planning capabilities of SWM, we start by applying a sampling-based planning method, MPPI, to a SWM model on LangTable and OGBench tasks. As shown in tab. 1, we are able to directly plan on top of our semantic world model using sampling-based planning methods, achieving close to perfect success rates on reaching and block separation tasks. However, the computational cost of the sampling-based planning method with large models makes it infeasible to run MPPI on more challenging tasks requiring a higher number of samples. Therefore, for more complicated tasks, we consider a

| Task | SWM |
|---|---|
| LT Reach Block | 100% |
| LT Separate Blocks | 100% |
| OG Reach Cube | 97% |

Table 1: **Planning Results** MPPI planning success rates over 100 seeds on LangTable and OG bench.

scenario in which a base policy generates a candidate trajectory that is refined using SWM and gradient-based optimization (described in Sec. 3.3.2). As shown in Fig. 5, our method improves substantially compared to baseline methods.

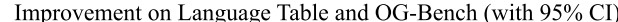

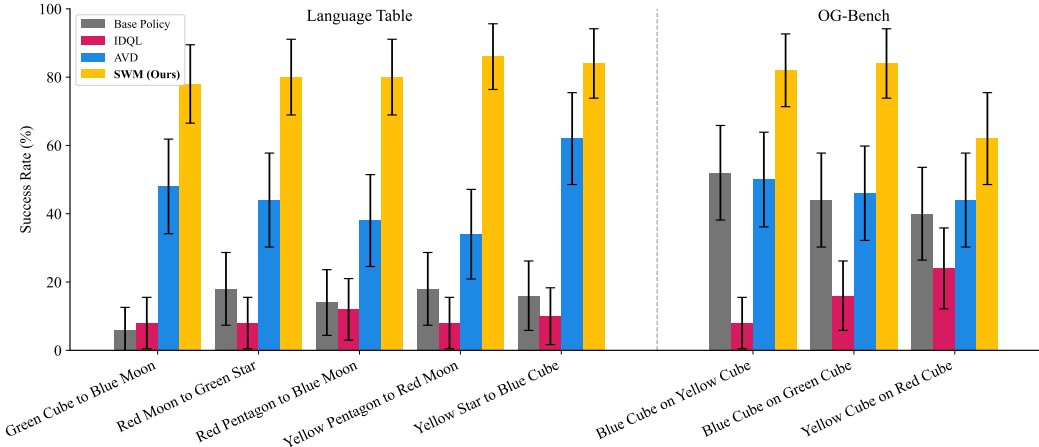

Figure 5: Improvement results across LangTable and OGBench

| Dataset Type | LangTable | | OGBench | |
|---|---|---|---|---|
| | Expert Data | Expert Data OOD | Expert Data | Expert Data OOD |
| Sub Optimal | $85.98 \pm 0.33$ | $81.99 \pm 1.46$ | $90.83 \pm 0.39$ | $85.56 \pm 1.10$ |
| Expert | $91.27 \pm 0.79$ | $86.49 \pm 0.39$ | $96.53 \pm 0.13$ | $87.33 \pm 2.13$ |
| Combined | $\mathbf{92.92} \pm 0.34$ | $\mathbf{88.32} \pm 2.10$ | $\mathbf{96.86} \pm 0.13$ | $\mathbf{88.16} \pm 1.54$ |

Table 2: **Future QA Performance.** Performance of future QA on test time expert datasets in both in-domain and out-of-domain block combinations. Reported standard deviation across 3 training seeds.

**Does suboptimal data improve modeling performance?** One of the key aspects of a world model is its ability to learn from suboptimal data. To measure the effects of suboptimal demonstrations, we create a test set of future QA data collected from expert demonstrations in both in-distribution and out-of-distribution environments. We then train models across three different seeds and fix hyperparameters to convergance with either the suboptimal data, optimal data, or a 50/50 mixed dataset. As seen in tab. 2, mixing in the suboptimal data improves accuracy over training on just expert data. SWM is also able to achieve moderate levels of performance by training only on suboptimal data, demonstrating how effective suboptimal data can be for training our world model.

**Does training preserve the generalization capabilities from the base VLM?** To measure the effects of VLM pretraining on generalization, we evaluate SWM on both compositional and scene out-of-distribution environments, depicted in Fig. 8. Since the offline dataset was misaligned with these evaluation tasks, we do not compare to the IDQL baseline.

To measure semantic compositional generalization, we introduce a new colored block and modify the existing block color-shape pairs in the LangTable environment. tab. 4 shows an average of $20.0\%$ improvement over the base policies under these conditions. This performance indicates that SWM is able to retain some of the pretraining knowledge, resulting in compositional generalization.

| Task | Base Policy | Video Diffusion | SWM (Ours) |
|---|---|---|---|
| MS1 | $6\% \pm 6.6$ | $8\% \pm 7.5$ | $\mathbf{50\%} \pm 13.9$ |
| MS2 | $4\% \pm 5.4$ | $2\% \pm 3.9$ | $\mathbf{66\%} \pm 13.1$ |
| MS3 | $4\% \pm 5.4$ | $2\% \pm 3.9$ | $\mathbf{54\%} \pm 13.8$ |
| MS4 | $2\% \pm 3.9$ | $4\% \pm 5.4$ | $\mathbf{54\%} \pm 13.8$ |

Table 3: **Multi-Step Results.** SWM model improvement results on four different multi-step compositional tasks. The tasks are as follows: MS1 - red pentagon to blue moon, yellow pentagon to red moon. MS2 - yellow star to blue cube, yellow pentagon to red moon. MS3 - yellow star to blue cube, red pentagon to blue moon. MS4 - green cube to blue moon, yellow pentagon to red moon. Reported success rates over $n = 50$ seeds with 95% confidence intervals (normal approximation).

| Task | Base Policy | AVD | SWM (Ours) |
|---|---|---|---|
| Push Blue Star to Red Cube | 54% ± 13.8 | 66% ± 13.1 | **86%** ± 9.6 |
| Push Yellow Moon to Purple Cube | 54% ± 13.8 | 56% ± 13.8 | **78%** ± 11.5 |
| Stack Red to Green OOD Background | 62% ± 13.5 | 28% ± 12.4 | **72%** ± 12.4 |
| Stack Blue to Yellow OOD Background | 50% ± 13.9 | 50% ± 13.9 | **70%** ± 12.7 |

Table 4: **Out-of-Distribution Improvement Results.** SWM model improvement results on tasks in LangTable and OG-Bench on out-of-distribution scenes. Reported success rates over $n = 50$ seeds with 95% confidence intervals (normal approximation). The highest mean per row is **bold**.

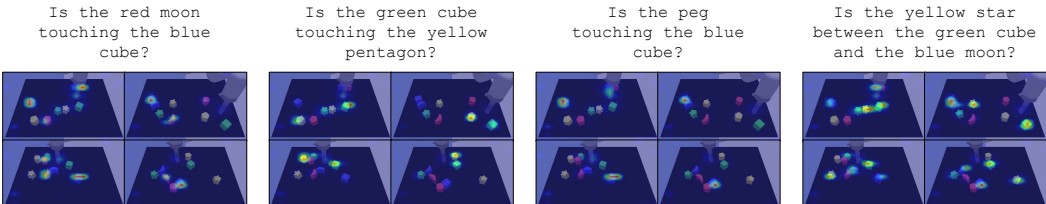

Figure 6: Visualization of the attention map from language tokens to image patches in the 4th transformer layer. The language tokens correctly attend to the task-relevant locations in the image depending on the prompt.

To test robustness to background changes, we change OGBench's background color to a novel combination. SWM is again able to demonstrate a $20\%$ boost in performance compared to the base policy and is able to generalize to these conditions, while the AVD method is unable to.

**Does the model's internal representations attend to the task-relevant information?** To understand the learned representations of the model, we visualize the attention maps from the language tokens to the image patches from an intermediate layer of the model. As shown in Fig. 6, the model correctly attends to the task-relevant location in the image depending on the language prompt. For example, when asked "Is the red moon touching the blue cube?", the attention score is higher on the image patches corresponding to the objects. Although we never finetuned on questions with more than two objects, we found the model to correctly attend to three objects when asked to. This shows that the model inherits generalization from the pretrained VLM. In sec. A.5.6 we provide more visualizations of individual layers as well as entire trajectories.

## 5 CONCLUSIONS

We present Semantic World Models, a novel world modeling approach that explicitly models future outcomes through future QA without needing to reconstruct or use pixel-level information as a training objective. We demonstrate that our approach can be used both with sampling-based planning methods and through the lens of policy improvement. We demonstrate considerable gains over pixel-based world modeling and offline RL methods, suggesting SWM could be the basis of a new framework for world modeling.

### 5.1 LIMITATIONS AND FUTURE WORK

While Semantic World Models demonstrate strong performance on multiple tasks, several limitations remain. First, the high parameter count of the base VLM makes sample-based planning methods too computationally expensive to perform on a single GPU or at a reasonable control frequency. The gradient-based planning method is significantly more efficient, but requires a base policy to propose the initial trajectory. Second, we also require ground truth simulation information in order to construct the SAQA dataset, which would be hard to get in real-world robotic environments.

This leads to some promising future directions to address these challenges. Instead of using PaliGemma as the base VLM, there is recent work towards training smaller VLMs, such as FastVLM or SmolVLM (Marafioti et al., 2025; Kumar et al., 2025). These smaller VLMs could enable sampling-based planning to scale up to more challenging tasks, thereby eliminating the need for

a base policy. We also believe it is a promising direction to replace the oracle-generated QA pairs with those directly derived from a base VLM model. This would enable scaling up both the diversity of data and the ability to include real data in the training recipe of a Semantic World Model.

## REPRODUCIBILITY

To promote reproducibility and facilitate building upon this work, we will release code and trained model weights to enable independent reproduction of our results. All of our reported results were obtained across multiple seeds, and we included multiple different goal configurations of each task to ensure reproducibility of our findings.

## GENERATIVE AI USAGE

LLM tools were used to refine the writing, and GitHub Copilot was used for code-writing assistance.

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

# A    APPENDIX

## A.1    MODEL ARCHITECTURE AND TRAINING DETAILS

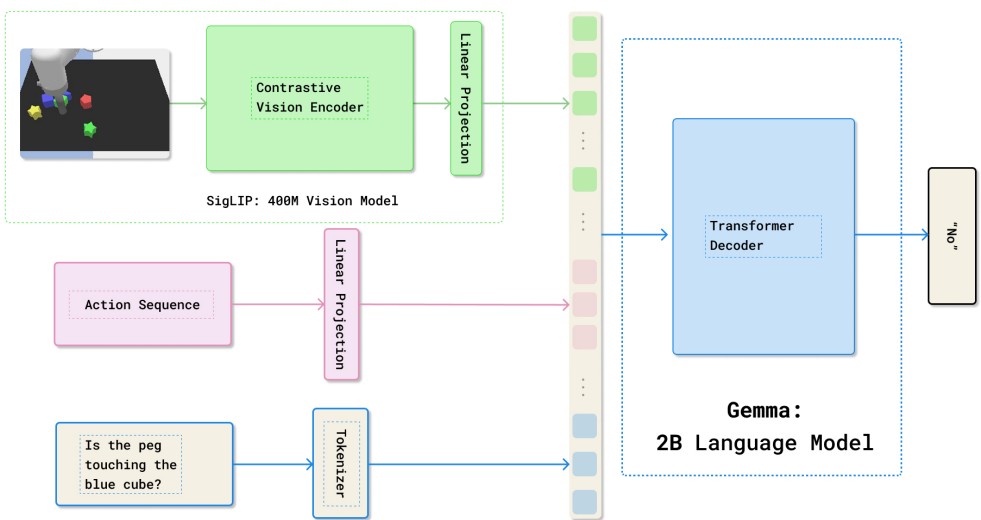

Figure 7: Architecture of Semantic World Model

Fig. 7 shows the architecture of Semantic World Model. We use the Paligemma 3B checkpoint as our base model. The only new component we introduce is a linear projection matrix that is dimension `act_dim×2048` where 2048 is the embed dimension of the Gemma model. We perform full weight fine-tuning on all model parameters using a linear LR decay starting at $1e^{-5}$ for approximately $24,000$ gradient steps on LangTable and $64,000$ gradient steps for OGBench. We use an effective batch size of 96. Each model is trained on a node comprising 4 AMD Instinct MI250X GPUs (each equipped with 2 MI200 GPU accelerators), resulting in a total training time of approximately 24 hours.

## A.2    BASELINES AND HYPERPARAMETERS

IDQL (Hansen-Estruch et al., 2023) is an offline RL method that applies implicit Q-learning to reweight a behavior diffusion-based policy. We use the base diffusion policy architecture for SWM as the policy for IDQL, except with an action horizon of 8 instead of 16. For the Q and Value functions in IDQL, we only condition on the current observation.

For the AVD baseline, we train a latent action-conditioned transformer video diffusion model, based on the architecture of Unified World Models (Zhu et al., 2025), without the action prediction head. Due to the computational cost of running the AVD forward and then using the generated frame for VQA, we are unable to run this baseline with a high number of samples. Since the MPPI initial samples were initialized from the base policy, we perform 10 iterations of MPPI with 16 samples to get our final action prediction. Each AVD run takes around 10 hours on a single GPU.

The hyperparameters used for the base diffusion model, the IDQL algorithm, and the AVD model are detailed in tab. 15. The only difference across environments is the size of the input image. All models are trained with the AdamW optimizer (Loshchilov & Hutter, 2019).

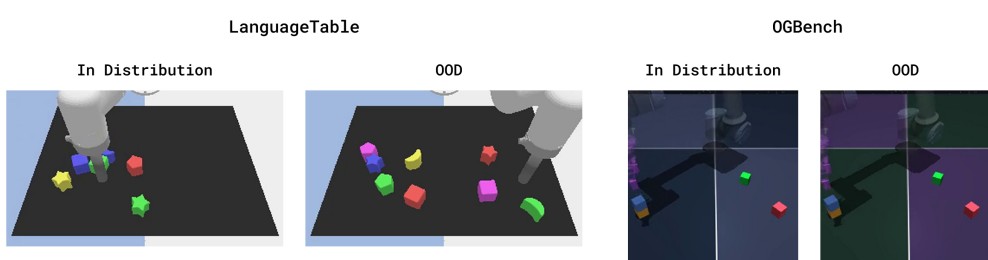

Figure 8: Out-of-distribution configurations for the evaluation tasks

## A.3 ENVIRONMENTS AND TASKS

### A.3.1 ENVIRONMENT DETAILS

Fig. 4 shows an example of each type of task we used to evaluate SWM. In Fig. 8, we provide examples of out-of-distribution configurations used to evaluate the generalization capabilities of SWM. More details about each environment and task are discussed below.

**LangTable** The LangTable environment has a control frequency of 10 Hz. For each task, we terminate each episode after 120 environment steps. Our observation space is a single $180 \times 320$ RGB image of the table. The action space is xy delta poses, ranging from -.03 to .03. Our reach block task is marked as a success if the peg made contact with the target block. The separate block task is marked as a success if the L2 distance between the target block and the blocks to separate it from is over .1 M. For pushing blocks together, the episode is marked as a success if the L2 distance between the two target blocks is less than .075. The expert and noisy demonstrations used for our offline dataset and expert diffusion dataset are collected on environment seeds 0-300, and we evaluate on seeds 6000-6050. For the SWM improvement, we use an action chunk of 8, a gradient learning rate of 0.02, 10 planning iterations, and execute 4 out of the 16 predicted actions before replanning. We use a gradient clipping of 1 before updating each action during planning.

**OGBench** We use the cube environment as the basis for our tasks. This environment has a control frequency of 10Hz, and we terminate each episode after 200 steps. Our observation space is a single $224 \times 244$ RGB image. The action space is 5-dimensional, comprising of delta xyz and orientation, and a gripper action. For the ReachCube task, we measure success as the gripper pads touching the cube. For our cube stacking task, we initialize all block poses randomly and then define success as the first cube being stacked on top of the second cube, with a gap between the top cube and the robotic gripper. The expert and noisy demonstrations used for our offline dataset and expert diffusion dataset are collected on environment seeds 0-300, and we evaluate on seeds 6000-6050. For the SWM improvement, we use an action chunk of 8, a gradient learning rate of 0.2, 20 planning iterations, and execute 4 out of the 16 predicted actions before replanning. We use gradient clipping of 10 before updating each action during planning.

### A.3.2 QUESTION-ANSWER DATASET CURATION

We precompute the future QA pairs for our offline dataset. For each state, we sample four different action horizon lengths between 0 and 20, and generate a set of questions for each sampled horizon. Tab. 5 shows the question types and an example of each question type on both the LangTable and OGBench environments.

For each question type, we also use multiple variations in wording. For example, for *block touching* questions, given two blocks {block1} and {block2}, we use:

- Is the {block1} touching the {block2}?
- Are the {block1} and {block2} blocks in contact with each other?
- Is there contact between the {block1} block and the {block2} block?
- Does the {block1} touch the {block2}?
- Is the {block1} block in physical contact with the {block2} block?

- Are the {block1} and {block2} blocks touching each other?
- Is the {block1} and {block2} directly touching?
- Do the {block1} and {block2} blocks meet?

### A.3.3 TASK SPECIFICATION

For each task, we use a fixed set of questions and answers to specify the goals. All of our tasks are single-subgoal tasks except the stack cube task, which has two goals. In order to create a multi-step task for LangTable, we use two subgoals of independent Block to Block tasks, and use the SWM to pick the behavior policy and the subgoal to use. The questions, answers, and weights for all tasks are shown in Tab. 5.

Table 5: QA pairs used for task rewards

| Task | Question | Weight | Desired Answer |
|------|----------|--------|----------------|
| Reaching LT | Is the robotic peg touching the {target_block}? | 0.8 | Yes |
| | Is the robotic peg closer to the {target_block}? | 0.2 | Yes |
| Reaching OG | Is the robotic gripper touching the {target_block}? | 0.8 | Yes |
| | Is the robotic gripper closer to the {target_block}? | 0.2 | Yes |
| Separate Blocks | Is the robotic peg touching the {center_block}? | 0.6 | Yes |
| | Is the {avoid block} touching the {center block}? | 0.4 | No |
| Block to Block | Is the {first_block} touching the {second_block}? | 0.8 | Yes |
| | Are the {first_block} and the {second_block} closer together? | 0.2 | Yes |
| Cube Stacking | **Subgoal 1: Pick up the first cube** | | |
| | Is the robot grasping the {first_block}? | 1.0 | Yes |
| | **Subgoal 2: Stack the blocks** | | |
| | Is the {first_block} on top of the {second_block}? | 0.6 | Yes |
| | Is the robot grasping the {first_block}? | 0.4 | Yes |

### A.4 FULL IMPROVEMENT RESULTS

We provide the full improvement results corresponding to Fig. 5 in the experiments section.

### A.5 ABLATIONS

### A.5.1 ABLATION ON QUESTION WEIGHTS

We ablate the inclusion of question weights across all of the in-distribution LangTable and OGBench tasks. We found that removing weights decreased performance by an average of $2.4\%$ in LangTable and increased performance by $3.3\%$ in ogbench. Full results are in Tab. 7.

### A.5.2 ABLATION ON PLANNING WITHOUT BASE POLICY

To evaluate whether SWM can plan effectively without relying on a base policy, we conducted an ablation study on three LangTable pushing tasks. In this setting, the initial action sequence was sampled uniformly from the environment's action space, removing any prior structure provided by a base policy. Despite the lack of a warm start, SWM was able to successfully plan under these randomly initialized trajectories, achieving success rates of $46\%$, $50\%$, and $58\%$ on the tasks of pushing the yellow pentagon to the red moon, pushing the red moon to the green star, and pushing the yellow star to the blue cube. These results demonstrate that the gradients are reasonable even from random initializations.

Table 6: **Improvement Results.** SWM model improvement results on planning tasks in LangTable and OG-Bench on in-distribution scenes. Reported success rates over $n = 50$ seeds with 95% confidence intervals (normal approximation). The top tasks are LangTable and the bottom tasks are OGBench.

| Task | Base Policy | IDQL | AVD | SWM |
|---|---|---|---|---|
| Push Green Cube to Blue Moon | $6\% \pm 6.6$ | $8\% \pm 7.5$ | $48\% \pm 13.8$ | $\textbf{78\%} \pm 11.5$ |
| Push Red Moon to Green Star | $18\% \pm 10.6$ | $8\% \pm 7.5$ | $44\% \pm 13.8$ | $\textbf{80\%} \pm 11.1$ |
| Push Red Pentagon to Blue Moon | $14\% \pm 9.6$ | $12\% \pm 9.0$ | $38\% \pm 13.5$ | $\textbf{80\%} \pm 11.1$ |
| Push Yellow Pentagon to Red Moon | $18\% \pm 10.6$ | $8\% \pm 7.5$ | $34\% \pm 13.1$ | $\textbf{86\%} \pm 9.6$ |
| Push Yellow Star to Blue Cube | $16\% \pm 10.2$ | $10\% \pm 8.3$ | $62\% \pm 13.5$ | $\textbf{84\%} \pm 10.2$ |
| Stack Blue Cube on Yellow Cube | $52\% \pm 13.8$ | $8\% \pm 7.5$ | $50\% \pm 13.9$ | $\textbf{82\%} \pm 10.6$ |
| Stack Blue Cube on Green Cube | $44\% \pm 13.8$ | $16\% \pm 10.2$ | $46\% \pm 13.8$ | $\textbf{84\%} \pm 10.2$ |
| Stack Yellow Cube on Red Cube | $40\% \pm 13.6$ | $24\% \pm 11.8$ | $44\% \pm 13.8$ | $\textbf{62\%} \pm 13.5$ |

### A.5.3 ABLATION ON QUESTION PHRASINGS

To evaluate the robustness of SWM to different question phrasings, we conducted an ablation measuring performance under both in-distribution and out-of-distribution question phrasings. For each task, we evaluated SWM using two different phrasings seen during training and two novel OOD phrasings not present in the SAQA dataset. As shown in Table 8, SWM maintains strong performance across all phrasing variants, with only minor drops under OOD formulations. These results demonstrate SWMs robustness to new question phrasings.

Table 7: **Ablation on question weights.** Success rates for SWM with weights vs. SWM without weights on LangTable and OGBench tasks. Reported over $n = 50$ seeds with 95% confidence intervals.

| Task | SWM (with weights) | SWM (no weights) |
|---|---|---|
| Push Green Cube to Blue Moon | $78\% \pm 11.5$ | $72\% \pm 12.4$ |
| Push Red Moon to Green Star | $80\% \pm 11.1$ | $78\% \pm 11.4$ |
| Push Red Pentagon to Blue Moon | $80\% \pm 11.1$ | $82\% \pm 10.6$ |
| Push Yellow Pentagon to Red Moon | $86\% \pm 9.6$ | $88\% \pm 9.0$ |
| Push Yellow Star to Blue Cube | $84\% \pm 10.2$ | $76\% \pm 11.8$ |
| Stack Blue Cube on Yellow Cube | $82\% \pm 10.6$ | $82\% \pm 10.6$ |
| Stack Blue Cube on Green Cube | $84\% \pm 10.2$ | $78\% \pm 11.4$ |
| Stack Yellow Cube on Red Cube | $62\% \pm 13.5$ | $78\% \pm 11.4$ |

Table 8: **Ablation on Question Phrasing.** Success rates of SWM under in-distribution (ID) and out-of-distribution (OOD) task phrasings. Reported with 95% confidence intervals.

| Task | Base Policy | ID 1 | ID 2 | OOD 1 | OOD 2 |
|---|---|---|---|---|---|
| Push Red Moon to Green Star | $18\% \pm 10.6$ | $72\%$ | $88\% \pm 9.0$ | $84\% \pm 10.2$ | $78\% \pm 11.5$ |
| Push Yellow Star to Blue Cube | $16\% \pm 10.2$ | $78\%$ | $86\% \pm 9.6$ | $84\% \pm 10.2$ | $72\% \pm 12.4$ |
| Push Yellow Pentagon to Red Moon | $18\% \pm 10.6$ | $88\%$ | $88\% \pm 9.0$ | $86\% \pm 9.6$ | $76\% \pm 11.8$ |

### A.5.4 AUTOMATIC QUESTION GENERATION

To validate the broader applicability of SWM beyond simulators with privileged state information, we evaluate an automatic dataset generation pipeline using a VLM to provide supervision for question answers. We evaluate the accuracy of Gemini Embodied Reasoning (Team, 2025) in providing question-answer supervision on both LangTable and OGBench. In addition, we have measured the accuracy on a small set of manually annotated question-answer pairs from the Droid datasets. Across both settings, Gemini achieved strong accuracy compared to ground-truth answers, indicating that

Table 9: Gemini-ER Accuracy on LangTable

| Question Type | Accuracy |
|---|---|
| block_touching | 0.93 |
| peg_to_block | 0.89 |
| block_closer | 0.88 |
| **Average** | **0.90** |

Table 10: Gemini-ER Accuracy on OGBench

| Question Type | Accuracy |
|---|---|
| cube_grasped | 1.00 |
| block_ontop_block | 0.95 |
| block_touching_block | 0.97 |
| block_block_closer | 0.75 |
| **Average** | **0.92** |

Table 11: Gemini-ER Accuracy on DROID

| Question Type | Accuracy |
|---|---|
| claw_hold | 0.89 |
| obj | 0.91 |
| obj_relative | 0.82 |
| **Average** | **0.87** |

Table 12: Success rate (%) on long-horizon LangTable tasks with and without LLM-derived task decompositions over 50 seeds. MS1 - red pentagon to blue moon, yellow pentagon to red moon. MS2 - yellow star to blue cube, yellow pentagon to red moon. MS3 - yellow star to blue cube, red pentagon to blue moon. MS4 - green cube to blue moon, yellow pentagon to red moon.

| Task | Base Policy | SWM+grad | SWM+grad + LLM planning |
|---|---|---|---|
| MS1 | 2 | 50 | 62 |
| MS2 | 4 | 66 | 42 |
| MS3 | 4 | 54 | 52 |
| MS4 | 6 | 54 | 44 |
| **Average** | **4** | **56** | **50** |

frontier VLMs show a path towards generating the SAQA dataset without relying on oracle information in both sim and real. Detailed results are in Tab. 9, 10, and 11. For each question, we issued multiple queries to the model, counted each response as a vote, and selected the answer with the highest vote count.

### A.5.5 AUTOMATIC TASK DECOMPOSITION

We evaluate the feasibility of using a VLM for automatic high-level decomposition. We find that for long-horizon tasks, GPT 5.1 (OpenAI, 2024) is able to break the task into substasks and create a set of questions and desired answers to plan with. We observe that GPT-generated questions were also more varied in phrasing. When paired with SWM planning, these planning parameters perform comparably to our oracle question-answer set planning results. Results are shown in Tab. 12 We used structured JSON output with a full prompt text of "You are a robotic agent planning to push blocks around on a table. Break your task down into key information and brief, absolute and relative questions. Question examples we've trained on: Is the red star touching the blue cube? Is the green cube next to the peg? Is the red star in the center of the board? Is the peg above the red cube block? Is the red star to the right of the blue cube? Did the red cube move left? Did the red star block move? Did the robotic peg move downward? Are the red star and blue cube closer together? Is the robotic peg closer to the red cube?"

### A.5.6 VISUALIZATION OF ATTENTION MAPS

We provide additional visualizations of the attention map. In Fig. 9, we visualize the average attention scores from language tokens to image tokens on a consecutive trajectory. We find that different layers capture different semantic information. For example, layers 4 and 6 attend to the red moon and the blue block, whereas later layers also attend to the peg, likely because of the need to

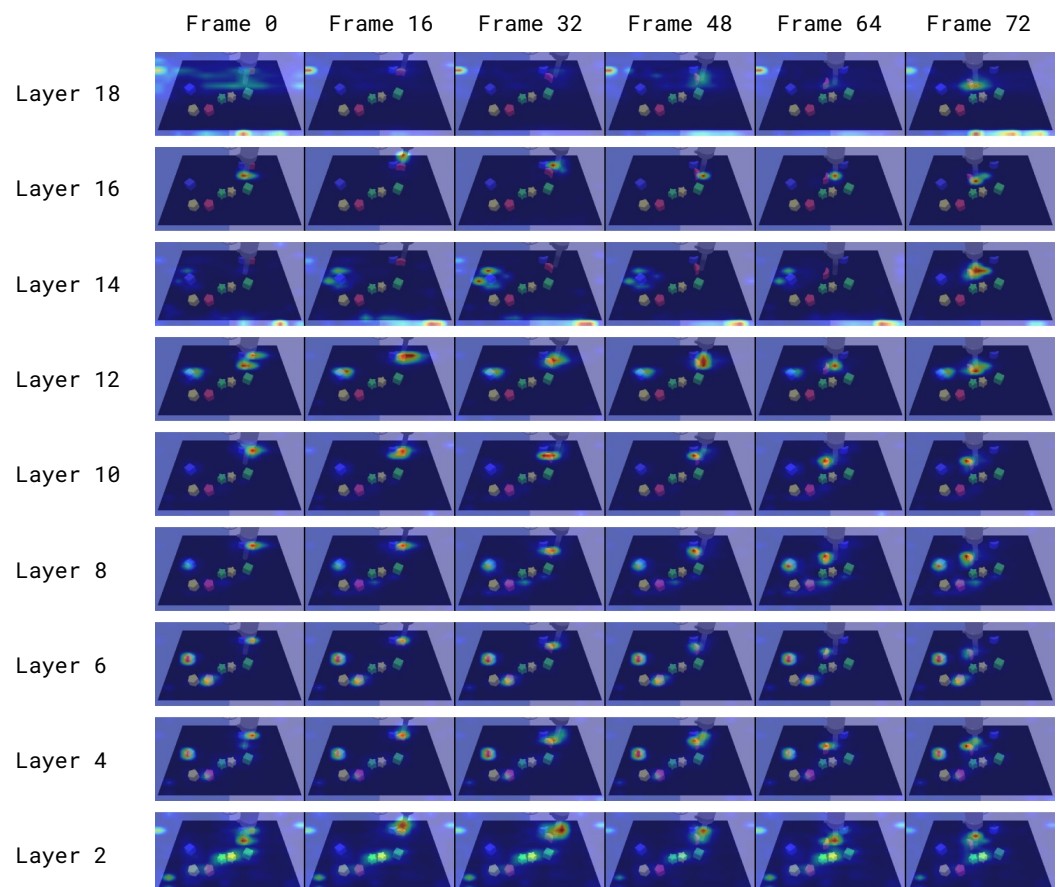

Figure 9: Attention maps in different layers of SWM . Question: "Is the red moon touching the blue block?"

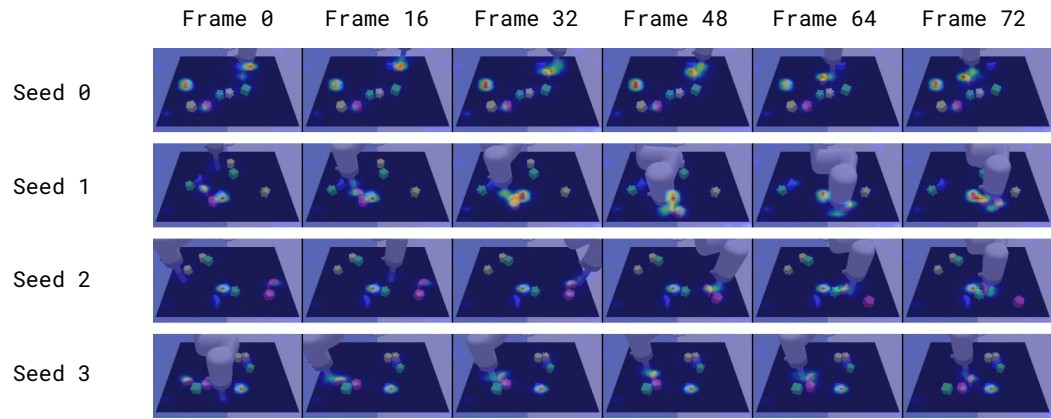

Figure 10: Attention maps for different trajectories. Question: "Is the red moon touching the blue block?"

reason about the result of actions. In Fig. 10 we visualize the attention map in layer 4 on different trajectories, showing that the layer consistently attends to the correct objects.

Figure 11: Visualization of gradient-based planning on the LangTable - Red Pentagon to Blue Moon task. The initially proposed action sequence is on the left, and updates to this action sequence go progressively to the right, approaching the optimal trajectory over successive gradient steps.

### A.5.7 VISUALIZATION OF GRADIENT-BASED PLANNING

We visualize the gradient-based planning procedure in Fig. 11. As planning iteration progresses, the candidate action sequence gradually extends to pushing the red pentagon to the blue moon, approaching the optimal trajectory over successive gradient steps.

### A.5.8 PLANNING EFFICENCY

We measure the effective environment Hz of AVD, MPPI, and our gradient-based method in LangTable. For our comparison, we fix the number of MPPI samples and number of MPPI planning steps to what we use in our AVD baseline, which is eight iterations with 16 samples. For gradient-based planning, we use the same parameters as those in the LangTable, specifically 10 iterations on a single candidate trajectory. For all three methods, we use a reward sub-chunk size of 8 and a horizon of 16. For the SWM gradient-based planning, we benchmarked the speed for a single forward and backwards pass on one action chunk of size 16 with one question. The forward pass for a single frame and action chunk takes on average 0.036 seconds, and the backward pass takes on average 0.0262 seconds. All numbers above are on a NVIDIA A100 GPU using bf16 precision. It is possible to run gradient-based improvement on a single RTX 4090.

Table 13: Planning speed comparison across different methods

| Method | Time per action chunk (Seconds) |
| --- | --- |
| AVD | 676.41 |
| MPPI | 4.48 |
| Gradient-based | 1.56 |

Table 14: Question types and examples for LangTable and OGBench

| Type | Example |
| --- | --- |
| **LangTable** | |
| Block touching | Is the red star touching the blue cube? |
| Peg to block | Is the green cube next to the peg? |
| Block board position | Is the red star in the center of the board? |
| Peg block relative direction | Is the peg above the red cube block? |
| Block to block relative direction | Is the red star to the right of the blue cube? |
| Block move direction | Did the red cube move left? |
| Block move | Did the red star block move? |
| Peg move direction | Did the robotic peg move downward? |
| Block to block closer | Are the red star and blue cube closer together? |
| Peg to block closer | Is the robotic peg closer to the red cube? |
| | |
| **OGBench** | |
| Cube grasped | Is the red cube grasped by the robot? |
| Gripper touching block | Is the blue cube touching the robot gripper? |

Table 15: Hyperparameters for IDQL, Diffusion, and AVD Model

| Diffusion | |
| --- | --- |
| Batch size | 128 |
| Epochs | 100 |
| Action horizon | 16 |
| Observation horizon | 2 |
| Diffusion iters | 100 |
| Eval diffusion iters | 10 |
| Traj end padding (steps) | 12 |
| **IDQL** | |
| Gradient steps | 250,000 |
| Batch size | 128 |
| IQL $\tau$ | 0.8 |
| Test time samples | 1000 |
| Temperature | 0.5 |
| Discount ($\gamma$) | 0.99 |
| Critic hidden dim | 256 |
| Critic learning rate | 0.0003 |
| Num layers | 3 |
| **AVD Model** | |
| Embed dim | 768 |
| Vision backbone | ViT-B/32 |
| Timestep embed dim | 512 |
| Latent patch shape | [2,2,2] |
| Num Transformer Layers | 12 |
| Num heads | 12 |
| Train steps | 1000 |
| Inference steps | 50 |
| Total steps | 100,000 |
| Global batch size | 288 |
| Learning rate | 1e-4 |
| Weight decay | 1e-6 |

Table 14 – continued from previous page

| Type | Example |
| --- | --- |
| Block touching block | Is the green cube touching the yellow cube? |
| Block on top of block | Is the red cube on top of the blue cube? |
| Gripper closer to block | Is the gripper closer to the green cube? |
| Block closer to block | Is the red cube closer to the blue cube? |

