# OpenReview forum: "Semantic World Models"
_ICLR.cc/2026/Conference — Submitted to ICLR 2026_

### Official Review · Reviewer_RSdH · 2025-10-26

**Soundness:** 3
**Presentation:** 3
**Contribution:** 3
**Rating:** 6
**Confidence:** 4

**Summary:**

The paper introduces Semantic World Models (SWM), which frame world modeling as predicting high-level semantic outcomes via future VQA instead of reconstructing pixels. The authors trained a VLM (PaliGemma-3B) to answer questions about the future given current observations and actions. Experiments on planning benchmarks (LangTable and OGBench) show clear gains over pixel-based and offline RL baselines.

**Strengths:**

- Predicting high-level task-relevant info instead of generating pixels makes a lot of sense for effective world model-based planning
- The methodology is simple, clear, and sound.
- The paper is well written and quite easy to follow.

**Weaknesses:**

- The translation from goals to QA pairs is manual. Trying some LLM prompting for this step would be interesting. Otherwise, I suggest acknowledging this point in *Limitations*. For the current benchmarks, manual translation may be acceptable, but it will become a bottleneck for more diverse scenarios in the future.

- I am unsure about SWM’s task generalization. Given a new task, do we always need to train a separate SWM? Some discussion and exploration of this would be valuable for future work that goes beyond LangTable and OGBench. For example, one additional experiment could be to fine-tune PaliGemma jointly on the SAQA samples from both domains and then compare the effectiveness.

- In Section 3, it would be nice to frame the Semantic World Model as a more general framework, with the current VLM + yes/no VQA setup presented as one particular instantiation. This is a minor point—the current style is clear and straightforward.

- One additional potential advantage to highlight over generative world models is computational efficiency. Including some brief quantitative results would be strong.

Suggested related works for better coverage:

- *Efficient Exploration and Discriminative World Model Learning with an Object-Centric Abstraction*: abstract world models for planning. There are strong connections between SWM and the semi-MDP frameworks reviewed there, which also provide examples of how to formalize SWM.
- *Planning with Reasoning using Vision Language World Model*: planning with a VLM-based world model. The idea of language-based abstraction is shared.
- *Predicate Invention for Bilevel Planning*: related to the proposed future-VQA setup—their predicates act as verifiers of desired states.

**Questions:**

See weakness

---

> ### Author Response · Authors · 2025-11-27
>
> We want to thank the reviewer, RSdH, for their constructive feedback. We address each individual point below.
>
>
> -  (W1) The translation from goals to QA pairs is manual.
>
>    We evaluate the feasibility of using a VLM for automatic translation from goals to QA pairs. We find that for long-horizon tasks, GPT 5.1 is able to break the task into substasks and create a set of questions and desired answers to plan with. We observe that GPT-generated questions were also more varied in phrasing. When paired with SWM planning, these planning parameters perform comparably to our oracle question-answer set planning results.
>
>    | Task | Base Policy | SWM+grad | SWM+grad+LLM planning |
>    | :--- | :--- | :--- | :-- |
>    | green_cube_blue_moon_yellow_pentagon_red_moon | 6% | 54% | 44% |
>    | yellow_star_blue_cube_yellow_pentagon_red_moon | 4% | 66% | 42% |
>    | yellow_star_blue_cube_red_pentagon_blue_moon | 4% | 54% | 52% |
>    | red_pentagon_blue_moon_yellow_pentagon_red_moon | 2% | 50% | 62% |
>    | Average | 4% | 56% | 50% |
>
> - (W2) Generalization to new tasks.
>
>   For our experiments, we trained a single SWM model per environment, as they can generalize across tasks (different initial configurations and goals). We did not explore cross-domain semantic world models in our work, but it is a strong potential direction for future work.
>
> - (W3) Framing SWM as a more general framework.
>
>   Indeed, the SWM framework allows for a more general set of predicates beyond yes/no questions, where the fitness of the trajectory can be evaluated as the likelihood of generating the entire answer. In this work, we use the binary questions as a proof-of-concept. We will explore this more general framework in future work.
>
> - (W4) Highlight computational efficiency
>
>   Although SWM with gradient is substantially more efficient, and we report planning speeds for both SWM and the generative world model in Appendix A.5.8,  we did not fully optimize the generative world model for efficiency, and instead, for image generation performance, so we do not think that our results clearly distinguish the gains in computational efficiency.
>
> - (W5) Related works.
>
>   We thank the reviewer for their suggestions of related works. We have edited our related works section to include those suggestions.

---

### Official Review · Reviewer_xpRa · 2025-10-26

**Soundness:** 2
**Presentation:** 2
**Contribution:** 2
**Rating:** 4
**Confidence:** 3

**Summary:**

The paper reframes world-model planning as answering semantic questions about the future. A Semantic World Model (SWM) is built by fine-tuning a VLM (PaliGemma-3B) to answer future-conditioned questions given an image, an action sequence, and a natural-language query. Actions are mapped into the LLM token space by a learned projection inputs concatenate image embeddings, action embeddings, and question tokens; training uses cross-entropy on the answer token.

For planning, each task is specified as a small set of questions, desired answers, and weights. The value of a candidate is the (weighted) likelihood that SWM produces the desired answers, optionally summed over sub-chunks to reward earlier completion (Eqs. (1)–(2)). The paper instantiates both MPPI (sampling-based; Eq. (3)) and a gradient-based improvement over a diffusion base policy.

At test time, the planner uses a fixed, per-task question set (with weights) to score sequences; Table 7 lists the question templates used for LangTable/OGBench. During training, future-QA labels are programmatically generated from oracle future state across multiple horizons to form SAQA supervision.

Empirically, MPPI on SWM attains near-perfect success on simpler tasks (e.g., 100% on LangTable reach/separate; 97% on OGBench reach). Gradient-based planning provides substantial policy improvement over IDQL and an action-conditioned video-diffusion world model (AVD) while being far faster in wall-clock than AVD.

**Strengths:**

- Originality — “future QA” as control objective. Defining task value via SWM answer likelihoods aligns the model objective with decision-making and avoids pixel prediction; the action-token projection turns a pretrained VLM into an action-conditioned model with minimal surgery.

- Quality — clear task specification and planners. Tasks are explicitly defined via question/answer/weight sets (Table 7), and the value function includes a sub-chunking mechanism to encourage earlier completion (Eqs. (1)–(2)). Both MPPI (with a softmax-weighted update; Eq. (3)) and gradient-based optimization are implemented.

- Clarity — datasets and questions are concrete. SAQA generation from oracle futures is described with multiple horizons and phrasings; Appendix A.3 enumerates question types and variations. Figures/tables make the setup legible.

- Significance — strong planning and efficiency results. MPPI planning achieves near-perfect success on simple tasks; gradient-based planning improves a diffusion base policy and is orders of magnitude faster than AVD in time per action chunk (1.56 s vs. 676 s; MPPI 4.48 s).

**Weaknesses:**

- Privileged-state supervision limits real-world portability. SAQA labels depend on oracle future state; acquiring comparable labels on hardware is challenging. A path to weaker supervision (pseudo-labels from frozen VLMs, success detectors) would strengthen applicability.

- Baseline coverage for latent world models is narrow. Comparisons focus on AVD (pixel-level) and IDQL; omitting modern latent world-model planners (e.g., Dreamer/TD-MPC-style) makes it hard to isolate the benefit of semantic QA vs. latent prediction, under matched wall-clock.

- Calibration and reward design are underexplored. The value function sums (log-)likelihoods of desired answers across sub-chunks; language-model probabilities can be miscalibrated. An ablation on temperature/label smoothing and sensitivity to the chunk size and weights would test robustness.

- Execution details for MPPI are implicit. The paper specifies the MPPI distribution update (Eq. (3)) and reports speeds “per action chunk,” but it does not explicitly state whether the controller executes the updated mean plan (common in MPPI) or the best sample; please clarify. (For the gradient-based planner, they do state “execute 4 of 16 actions and replan.”)

- Compute trade-offs and reliance on a base policy. MPPI with large models is computationally heavy; the main gains on harder tasks come from gradient-based refinement of a diffusion base policy. Reporting matched wall-clock performance “from scratch” would sharpen the picture.

- Semantic scope is mostly binary. Table 7 shows binary QA templates; a few non-binary or numeric queries (counts, distances) would demonstrate broader semantic control.

**Questions:**

- Task questions at test time. Confirm that, for each task, you use the fixed per-task question set (with weights) in Table 7 to score candidates. Have you tried automatically deriving questions from a natural-language instruction at test time?

- MPPI action execution. Given Eq. (3), do you execute the updated mean sequence (standard MPPI) or the best-scoring sample in the receding-horizon loop? Please describe the exact selection rule used during roll-out.

- Calibration & value design. How calibrated are answer probabilities? Any temperature scaling or label-smoothing results? Sensitivity of performance to sub-chunk size and weights in Eqs. (1)–(2)?

- Weaker supervision for SAQA. Any preliminary results on generating future-QA labels without oracle state (e.g., pseudo-labels from a frozen VLM on future frames, or success detectors)?

- Baselines under compute parity. Can you include a compact comparison against a latent world-model planner (e.g., Dreamer or TD-MPC-style) under matched wall-clock and horizon?

- From-scratch planning on complex tasks. Beyond the simple cases where MPPI succeeds, can SWM plan without a base policy on multi-step or harder OGBench tasks under moderate compute? If not, what dominates the failure (search, query design, calibration)?

- Comparison to automatic question discovery: Work like https://arxiv.org/abs/2410.23156 and https://arxiv.org/abs/2501.00296 seems highly relevant where they automatically find task-relevant questions and use it for planning. Could you add a comparison to these approaches, clarifying similarities/differences in how questions are generated, how they interface with the planner, and the supervision required?

---

> ### Author Response · Authors · 2025-11-27
> **Response (1/2)**
>
> We want to thank the reviewer, xpRa, for their constructive feedback. We address each individual point below.
>
> - (W1, Q4) SAQA Privileged state supervision.
>
>   To validate the broader applicability of SWM beyond simulators with privileged state information, we evaluate an automatic dataset generation pipeline using a VLM to label the answers. We evaluate the accuracy of Gemini Embodied Reasoning on (1) the two simulated datasets used in our experiments (LangTable, OGBench) and (2) a small set of manually annotated question-answer pairs from the Droid datasets, a total of 500 questions across 50 scenes. Across both settings, Gemini achieved strong agreement with ground-truth answers, indicating that frontier VLMs show a path towards generating the SAQA dataset without relying on oracle information in both sim and real.
>
>   | LangTable | gemini-robotics-er-1.5-preview |
>   | :--- | :--- |
>   | block_touching | 93% |
>   | peg_to_block | 89% |
>   | block_closer | 88% |
>   | **accuracy** | **90%** |
>
>   | OGBench | gemini-robotics-er-1.5-preview |
>   | :--- | :--- |
>   | cube_grasped | 100% |
>   | block_ontop_block | 95% |
>   | block_touching_block | 97% |
>   | block_block_closer | 75% |
>   | **accuracy** | **92%** |
>
>   | DROID | gemini-robotics-er-1.5-preview |
>   | :--- | :--- |
>   | gripper_holding | 89% |
>   | obj |  91% |
>   | obj_relative | 82% |
>   | **accuracy** | **87%** |
>
> - (W2, Q5) Baseline coverage for latent world models is narrow.
>
>   To address a lack of a latent world model baseline, we evaluated Dreamer on the LangTable benchmark. We found that in the offline sparse reward setting, Dreamer was unable to solve our langtable tasks, achieving 1.33% on average. We further ran Dreamer in an online fixed-block initial-position setting with dense rewards, achieving 100% task success.  This contrast underscores Dreamer’s dependence on active exploration data, and demonstrates the effectiveness of SWM over latent world modeling approaches.
>
> - (W3, Q3) Calibration and reward design are underexplored.
>
>   We conduct additional ablation experiments on the robustness of SWM reward calibration by ablating both question weights and phrasings. As shown in the following tables, phrasing and question weights have a marginal impact on SWM’s performance. In all of our experiments with weights, we set the weights using heuristics.
>
>   | Task | Base Policy | SWM with Weights | SWM no weights | ID different phrasing | ood 1 phrasing | OOD 2 phrasing |
>   | :--- | :--- | :--- | :--- | :--- | :--- | :--- |
>   | Push Red moon green star | 18% | 78%  | 72% | 88% | 84% | 78% |
>   | Push Yellow Star to Blue Cube | 16% | 80% | 78% | 86% | 84% | 72% |
>   | Push Yellow Pentagon to Red Moon | 18%  | 86%  | 88% | 88% | 86% | 76% |
>
>
>   | Task | Base Policy | AVD | SWM weights | SWM no weights |
>   | :--- | :--- | :--- | :--- | :--- |
>   | Push Blue Star to Red Cube | 54%    | 66% | 86%  | 80% |
>   | Push Yellow Moon to Purple Cube | 54%  | 56% | 78% | 82% |
>   | Stack Red to Green OOD Background | 62% | 28% | 72% | 66% |
>   | Stack Blue to Yellow OOD Background | 50%  | 50% | 70%  | 70% |
>
>
>
> - (W4, Q2) Execution details for MPPI are implicit.
>
>   We roll out the mean sequence of the last iteration when using MPPI. We have updated the planning section of our paper with the explicit details of our selection rule.
> - (W5, Q6) Reliance on base policy
>
>   We run gradient-based planning with a random trajectory initialization. As shown below, gradient-based planning from scratch is prone to local minima. Hence, it is preferable to initialize gradient-based planning from a reasonable base policy.
>
>   | Task | SWM+grad from base policy | SWM+grad from scratch |
>   | :--- | :--- | :--- |
>   | yellow_pentagon_red_moon | 86% | 46% |
>   | red_moon_green_star | 80% | 50% |
>   | yellow_star_blue_cube | 84% | 58% |
>
> - (W6) Binary semantic scope
>
>   We thank the reviewer for bringing up non-binary semantic scope. Indeed, the SWM framework allows for a more general set of predicates beyond yes/no questions, where the fitness of the trajectory can be evaluated as the likelihood of generating the entire answer. In this work, we use the binary questions as a proof-of-concept. We will explore this more general framework in future work.

---

> > ### Author Response · Authors · 2025-11-27
> > **Response (2/2)**
> >
> > - (Q1) Have you tried automatically deriving questions from a natural-language instruction at test time?
> >
> >   The results with weights use the fixed questions and weights shown in the paper. We evaluate the feasibility of using a VLM for automatic high-level decomposition. We find that for long-horizon tasks, GPT 5.1 is able to break the task into substasks and create a set of questions and desired answers to plan with. We observe that GPT-generated questions were also more varied in phrasing. When paired with SWM planning, these planning parameters perform comparably to our oracle question-answer set planning results.
> >
> >   | Task | Base Policy | SWM+grad | SWM+grad+LLM planning |
> >   | :--- | :--- | :--- | :-- |
> >   | green_cube_blue_moon_yellow_pentagon_red_moon | 6% | 54% | 44% |
> >   | yellow_star_blue_cube_yellow_pentagon_red_moon | 4% | 66% | 42% |
> >   | yellow_star_blue_cube_red_pentagon_blue_moon | 4% | 54% | 52% |
> >   | red_pentagon_blue_moon_yellow_pentagon_red_moon | 2% | 50% | 62% |
> >   | Average | 4% | 56% | 50% |
> >
> > - (Q7) Comparison to automatic question discovery.
> >
> >   We thank the reviewer for pointing out the connection to automatic question or predicate discovery methods, such as “From Pixels to Predicates: Learning Symbolic World Models via Pretrained Vision-Language Models.” These approaches focus on deriving symbolic task decompositions or predicate chains to organize high-level reasoning and task decomposition. In contrast, our work targets a complementary task: given a task specification, SWM provides a mechanism for planning low-level actions to accomplish that task. We agree that automatic question/predicate discovery could serve as a valuable upstream component for SWM, automatically decomposing high-level goals into subtasks and question-answer pairs. We have added this discussion to the related work section to clarify the relationship and potential synergy between these approaches.

---

### Official Review · Reviewer_DSGT · 2025-10-31

**Soundness:** 2
**Presentation:** 3
**Contribution:** 2
**Rating:** 4
**Confidence:** 4

**Summary:**

This paper reframes world modeling as predicting task-relevant semantics rather than pixels, casting it as a visual question answering problem about future scenes. This allows the use of vision-language models (VLMs) as “semantic world models,” trained via supervised finetuning on image–action–text data. These models support planning and policy improvement with stronger generalization and robustness than reconstruction-based methods.

**Strengths:**

1. New framework: The core idea of shifting from pixel-level prediction to semantic, question-based prediction is novel and compelling. It directly addresses a known weakness of many video-based world models

2. Empirical results: The SWM achieves impressive results (Figure 5, Table 8), significantly outperforming the base policy and other baselines (IDQL, AVD).

3. Effective use of suboptimal data: The paper shows (Table 2) that model performance improves when trained on a combination of expert and suboptimal data, compared to using expert data alone.

**Weaknesses:**

1. The paper posits that a VLM could be used to decompose a high-level goal into these QA pairs (Section 2), but this is not demonstrated. The method requires a human to meticulously design a "curriculum" of questions to define a task, which is not scalable.

2. Comparisons: The comparison to the "Action Conditioned Video Diffusion" (AVD) baseline is not a fair "apples-to-apples" comparison. The AVD model is used to predict a future frame, and then the authors' own SWM model is used to perform VQA on that predicted frame to get a reward. This setup unfairly benefits the SWM's semantic reward structure. A more appropriate baseline would be to use a standard pixel-based model (like Dreamer or PlaNet) with a standard planning algorithm (like CEM) optimizing a task-specific reward (e.g., L2 distance to goal), rather than forcing it into the SWM's VQA-based reward framework. Furthermore, the reported planning time for AVD (676 seconds, Table 9) is so high it suggests a non-optimized implementation, further calling the comparison into question.

3. Lack of real-world application: The entire evaluation is conducted in simulation (LangTable, OGBench). The paper makes broad claims about a "powerful paradigm for robotic control" but provides zero evidence of the method's feasibility in the real world.

**Questions:**

All of my qeustions are listed in the weakness section. If my concerns are well addressed, I will raise my rating.

---

> ### Author Response · Authors · 2025-11-27
>
> We want to thank the reviewer, DSGT, for their constructive feedback. We address each individual point below.
>
> - (W1) Lack of VLM for high-level decomposition
>
>   We evaluate the feasibility of using a VLM for automatic high-level decomposition. We find that for long-horizon tasks, GPT 5.1 is able to break the task into substasks and create a set of questions and desired answers to plan with. We observe that GPT-generated questions were also more varied in phrasing. When paired with SWM planning, these planning parameters perform comparably to our oracle question-answer set planning results. We added this to section A.5.5 of the paper.
>
>   | Task | Base Policy | SWM+grad | SWM+grad+LLM planning |
>   | :--- | :--- | :--- | :--- |
>   | green_cube_blue_moon_yellow_pentagon_red_moon | 6% | 54% | 44% |
>   | yellow_star_blue_cube_yellow_pentagon_red_moon | 4% | 66% | 42% |
>   | yellow_star_blue_cube_red_pentagon_blue_moon | 4% | 54% | 52% |
>   | red_pentagon_blue_moon_yellow_pentagon_red_moon | 2% | 50% | 62% |
>   | Average | 4% | 56% | 50% |
>
>
> - (W2) Unfairness of AVD baseline and addition of standard pixel-based baseline.
>
>   While the AVD baseline is not apples to apples in terms of computation (as pointed out it takes 676 seconds per planning iteration), we used it to demonstrate the benefits in planning performance of not representing the entire future frame. We tuned the AVD baseline for image accuracy (100 evaluation diffusion iterations and a larger model size) rather than for computational efficiency to create the fairest comparison along this dimension.
>
>   To address a lack of a standard model-based RL baseline, we evaluated Dreamer on the LangTable benchmark. We found that in the offline sparse-reward setting, Dreamer was unable to solve our langtable tasks, achieving an average of 1.33%. We further ran Dreamer in an online fixed-block initial-position setting with dense rewards, achieving 100% task success.  This contrast underscores Dreamer’s dependence on active exploration data, and demonstrates the difficulty of the LangTable environment and the effectiveness of SWM.
>
> - (W3) Real-world feasibility
>
>   The main challenge for deploying SWM in the real world is the need of privileged state information to create the SAQA dataset. This assumption can be relaxed by using a VLM to label the dataset automatically. To validate this prospect, we evaluated the accuracy of Gemini Embodied 1.5 on (1) the two simulated datasets used in our experiments (LangTable, OGBench) and (2) a small set of manually annotated question-answer pairs from the Droid datasets, a total of 500 questions across 50 scenes. Across both settings, Gemini achieved strong agreement with ground-truth answers, indicating that frontier VLMs show a path towards generating the SAQA dataset without relying on oracle information in both sim and real.
>
>   | LangTable | gemini-robotics-er-1.5-preview |
>   | :--- | :--- |
>   | block_touching | 93% |
>   | peg_to_block | 89% |
>   | block_closer | 88% |
>   | **accuracy** | **90%** |
>
>   | OGBench | gemini-robotics-er-1.5-preview |
>   | :--- | :--- |
>   | cube_grasped | 100% |
>   | block_ontop_block | 95% |
>   | block_touching_block | 97% |
>   | block_block_closer | 75% |
>   | **accuracy** | **92%** |
>
>   | DROID | gemini-robotics-er-1.5-preview |
>   | :--- | :--- |
>   | gripper_holding | 89% |
>   | obj |  91% |
>   | obj_relative | 82% |
>   | **accuracy** | **87%** |

---

### Official Review · Reviewer_JTnb · 2025-11-01

**Soundness:** 3
**Presentation:** 3
**Contribution:** 3
**Rating:** 6
**Confidence:** 3

**Summary:**

The paper proposes Semantic World Models (SWM), reframing world modeling as future-tense visual QA instead of future pixel/latent prediction. A pre-trained VLM is adapted to answer questions about the outcomes of candidate action sequences; these answer likelihoods are aggregated to score trajectories. A simulation-generated SAQA dataset provides future Q/A labels (using privileged state). SWM is used for planning via (i) MPPI sampling and (ii) gradient-based refinement atop a base policy. On LangTable and OGBench, SWM improves task success over a pixel-diffusion world model and an offline RL baseline, shows some OOD robustness, and benefits from mixed-quality data.

**Strengths:**

- Novel formulation: The paper introduces a novel conceptual framing of world models as future-tense semantic predictors rather than pixel predictors. This reframing is non-trivial and departs meaningfully from both pixel/latent-based world modeling and language-conditioned policy architectures (e.g., VLAs), offering a new axis for research in decision-making with foundation models.

- Planning experiments: Compatible with both sampling and gradient refinement; the latter gives strong policy-improvement behavior.

- Empirical signal: Consistent gains over a pixel-diffusion world model and an offline RL baseline; robustness in some OOD settings; positive results with mixed-quality data.

- Clarity: The paper is generally well-written with helpful diagrams and ablations that support the core message.

**Weaknesses:**

- SAQA depends on privileged state: The central data engine uses oracle simulation state to label future Q/A tuples; there’s no empirical demonstration of a real-data pipeline (weak/self-supervised QA labels, multi-view, proprio/contact cues, etc.). This is a potential barrier for training this method on real-world data and further deployment and, IMO, the main limitation.

- From-scratch planning is not yet practical: Sampling with a large VLM is slow; the most effective mode is gradient refinement seeded by a competent base policy. The method thus acts as a policy-improvement operator rather than a standalone planner on harder tasks.

- Limited scope of evaluation: Simulated tabletop tasks with relatively short horizons; no real-robot trials and limited analysis of long-horizon multi-object goals.

- Baselines could broaden: Missing comparisons to language-space/CoT-style planners or semantic reward models trained directly from pixels; these would clarify where SWM’s semantic scoring helps most.

- Under-reported engineering details: No clear compute/latency budgets for MPPI vs gradient planning (per-timestep wall-clock, VRAM, #VLM passes), and limited sensitivity analyses (prompt phrasing, chunk length, weight selection).

**Questions:**

1) Real-world SAQA without privileged state: How do you envision generating future QA labels on real robot data where ground-truth object state is unavailable? (e.g., weak labels, VLM consistency, multi-view cues)

2) Planning compute & latency: What is the per-timestep compute cost for MPPI and gradient refinement (forward/backward passes, approximate ms per step)? This will clarify practical control rates.

3) Prompt / language robustness: How sensitive is planning performance to rephrasing of the questions at test time (synonyms, negations, multi-object phrasing)?

4) Dependence on base policy: Can the gradient-based planner make meaningful progress with a weaker or absent base policy, or is a strong initialization required?

5) Task spec & weights: How sensitive is performance to the manually chosen question weights, and do you see a viable path to automatically learning or deriving them?

---

> ### Author Response · Authors · 2025-11-27
>
> We want to thank reviewer JTnb for their constructive feedback. We address each individual point below.
> - (w1, Q1) SAQA dataset dependency on privileged state
>
>   To validate the broader applicability of SWM beyond simulators with privileged state information, we evaluate an automatic dataset generation pipeline using a VLM to label the answers. We evaluate the accuracy of Gemini Embodied Reasoning on (1) the two simulated datasets used in our experiments (LangTable, OGBench) and (2) a small set of manually annotated question-answer pairs from the Droid datasets, a total of 500 questions across 50 scenes. Across both settings, Gemini achieved strong agreement with ground-truth answers, indicating that frontier VLMs show a path towards generating the SAQA dataset without relying on oracle information in both sim and real.
>   | LangTable | gemini-robotics-er-1.5-preview |
>   | :--- | :--- |
>   | block_touching | 93% |
>   | peg_to_block | 89% |
>   | block_closer | 88% |
>   | **accuracy** | **90%** |
>
>   | OGBench | gemini-robotics-er-1.5-preview |
>   | :--- | :--- |
>   | cube_grasped | 100% |
>   | block_ontop_block | 95% |
>   | block_touching_block | 97% |
>   | block_block_closer | 75% |
>   | **accuracy** | **92%** |
>
>   | DROID | gemini-robotics-er-1.5-preview |
>   | :--- | :--- |
>   | gripper_holding | 89% |
>   | obj |  91% |
>   | obj_relative | 82% |
>   | **accuracy** | **87%** |
> - (w2, Q4) Planning dependency on base policy
>
>   We introduce the base policy because gradient-based planning is prone to local minima, as illustrated in the table below. While the size of the VLM renders sampling-based planning impractical on challenging tasks, we believe that our method scales with compute, and will become more feasible as compute becomes more efficient.
>   | Task | SWM+grad from base policy | SWM+grad from scratch |
>   | :--- | :--- | :--- |
>   | yellow_pentagon_red_moon | 86% | 46% |
>   | red_moon_green_star | 80% | 50% |
>   | yellow_star_blue_cube | 84% | 58% |
> - (W4) Baselines could broaden
>
>   We baselined against a standard latent world modeling approach, Dreamer. We found that in the offline sparse reward setting, Dreamer was unable to solve our langtable tasks, achieving 1.33% on average. We further ran Dreamer in an online fixed-block initial-position setting with dense rewards, achieving 100% task success.  This contrast underscores Dreamer’s dependence on active exploration data, and demonstrates the difficulty of the LangTable environment and the effectiveness of SWM.
>
> - (W5, Q2) Under-reported engineering details
>
>   We refer to Appendix A5.8 for information on full-step planning times. Specifically, the forward pass for a single frame and action chunk takes on average 0.036 seconds, and the backward pass takes on average 0.0262 seconds. We revised our Appendix A.5.8 to include these numbers and to clarify our engineering details.
>
> - (Q3, Q5) Prompt/language robustness, sensitivity to chosen question weights
>   We conduct additional ablation experiments on the robustness of SWM to prompts and question weights. As shown in the following tables, phrasing and question weights have a marginal impact on SWM’s performance. In our experiments, we set them using heuristics.
>
>   | Task | Base Policy | SWM with Weights | SWM no weights | ID different phrasing | ood 1 phrasing | OOD 2 phrasing |
>   | :--- | :--- | :--- | :--- | :--- | :--- | :--- |
>   | Push Red moon green star | 18% | 78%  | 72% | 88% | 84% | 78% |
>   | Push Yellow Star to Blue Cube | 16% | 80% | 78% | 86% | 84% | 72% |
>   | Push Yellow Pentagon to Red Moon | 18%  | 86%  | 88% | 88% | 86% | 76% |
>
>
>   | Task | Base Policy | AVD | SWM weights | SWM no weights |
>   | :--- | :--- | :--- | :--- | :--- |
>   | Push Blue Star to Red Cube | 54%    | 66% | 86%  | 80% |
>   | Push Yellow Moon to Purple Cube | 54%  | 56% | 78% | 82% |
>   | Stack Red to Green OOD Background | 62% | 28% | 72% | 66% |
>   | Stack Blue to Yellow OOD Background | 50%  | 50% | 70%  | 70% |

---

### Author Response · Authors · 2025-11-27
**General Response**

We thank all reviewers for their time and feedback on our paper. We summarize the new results here and defer the details to the individual responses.
1. Automatic SAQA dataset generation (JTnb, xpRa): To validate the broader applicability of SWM beyond simulators with oracle state information, we evaluate an automated dataset generation pipeline. We use the Gemini Embodied Reasoning model to automatically generate answers to questions about the future. We find the pipeline to achieve high validation accuracy on both simulated dataset (LangTable) and real-world datasets (DROID).  We added these results to section A.5.4 in our paper.
2. Automatic task decomposition (RSdH, xpRa): We evaluate using GPT 5.1 to automatically generate the questions and desired answers for a high-level goal. This reduces the effort required to apply SWM to a new task. We added these results to section A.5.5 in our paper.
3. Latent Model-Based RL baseline (DSGT, xpRa, JTnb): We evaluate a latent model-based RL method, Dreamer, on the LangTable tasks. Across three offline tasks, we find that it is unable to solve them, achieving an average success rate of 1.33% from the same offline dataset and sparse rewards used for our SWM, IDQL, and AVD methods. We validated our results using the online version of Dreamer with fixed initialization and dense rewards. This variant achieves 100% success, indicating Dreamers' reliance on online exploration data and dense rewards.
4. Additional ablations
  - Gradient-based planning from scratch (JTnb, xpRa): we try gradient-based planning from randomly initialized trajectories instead of a base policy. We get lower success rates because of local minima. We added these results to section A.5.2 of our paper.
  - Effectiveness of question weights and phrasing (JTnb, xpRa): we ablate the question weights and phrasing. We found that removing the question weights or changing the phrasing had a marginal impact on success rates. This shows that SWM does not depend strongly on the question weights or phrasing. We added these to section A.5.3 in the paper.

---

### Meta-Review · Area_Chair_Zmh1 · 2026-01-06

**Summary:**

A semantic world model is proposed, by fine tuning a VLM to predict answers to pre-specified questions conditioned on action sequences.

**Reviewer Concerns:**

All reviewers acknowledged the novelty of the conceptual idea - this seems like a natural next step for planning with world models, and the proposed approach is simple and elegant.

Several concerns were raised about the technical capabilities of the approach, mainly:
1. The requirement to have ground state when creating the data for the SWM.
2. The requirement to design the set of questions/answers manually.
3. The evaluation on relatively simple domains.
4. Additional baselines like dreamer.

The authors tried to address (1) by showing that the Gemini Embodied Reasoning model provides relatively similar answers to GT on a chosen set of domains. In my opinion this don't fully address this concern (we don't really know how 89.67% agreement affect the overall performance, and how general this is).

The authors tried to address (2) by showing results with GPT-generated decompositions. In my opinion this is an ad-hoc response that don't fully address this concern (we do not understand what the limitation of this approach is - when do we expect ChatGPT to work?)

The authors fully addressed (4) by adding comparisons with dreamer.

Concern (3) remains, but I did not give it much weight due to the conceptual contribution of the work.

An additional concern that I found when reading the paper was the comparison with the AVD baseline. Beyond the concerns that Reviewer DSGT raised on this topic, it is not clear to me how the SWM was evaluated in this baseline, since it is not trained for inputs of this kind (future predicted frame).
In my opinion, the comparison with AVD baseline should be given much more weight and explanations - this is **the major** claim in the paper - that semantic models are better than pixel reconstruction, and the current results in this matter are not conving/rigorous enough.

**Reviewer Scores:**

6,4,4,6

My decision is based on the reviews, the rebuttal, and my own careful evaluation of the paper.
This is a borderline paper, as reflected in the scores. It makes big "conceptual" claims, but the experimental validation is not convincing enough to back them up. It is hard to predict how the reviewers would have changed their scores, but the bottom line is that several major concerns remain after the rebuttal.

It may definitely be that with another round of discussion, additional concerns would have been resolved. I therefore encourage the authors to resubmit this work after addressing the concerns, focusing on the major issues above.

---

### Decision · Program_Chairs · 2026-01-26

Reject